# Efficacy of Liquid Embolic Agent Treatment in Hemorrhagic Peripheral Intracranial Aneurysms: A Single-Center Experience

**DOI:** 10.3390/brainsci12091264

**Published:** 2022-09-19

**Authors:** Zong Zhuang, Qi Zhu, Xun-Zhi Liu, Hai-Ping Ling, Shi-Jie Na, Tao Liu, Yu-Hua Zhang, Chun-Hua Hang, Kai-Dong Liu, Qing-Rong Zhang

**Affiliations:** 1Department of Neurosurgery, The Affiliated Drum Tower Hospital, School of Medicine, Nanjing University, Zhongshan Road 321, Nanjing 210008, China; 2Department of Neurosurgery, Nanjing Drum Tower Hospital Clinical College of Nanjing Medical University, Nanjing 210008, China

**Keywords:** peripheral intracranial aneurysm, brain hemorrhage, liquid embolization agents, interventional therapy

## Abstract

*Objective:* To evaluate the efficacy of liquid embolization agents for treating various hemorrhagic peripheral intracranial aneurysms. *Methods**:* We retrospectively analyzed 38 patients who suffered from hemorrhagic peripheral intracranial aneurysms and were treated with liquid embolization agents. We used the modified Rankin scale for follow-up at 6 months postoperatively, and digital subtraction angiography follow-up was performed 6 months postoperatively. *Results**:* Of the 38 patients (ten of simple peripheral intracranial aneurysms, six of Moyamoya disease (MMD), and 22 of arteriovenous malformation (AVM)), posterior circulation accounted for the most significant proportion (57.9%), followed by anterior circulation (21.1%) and intranidal aneurysms (21.1%). Intraoperative hemorrhage occurred in four cases, postoperative cerebral infarction occurred in four cases, two patients encountered microcatheter retention, and intraoperative thrombosis took place in the basilar artery of a patient with an arteriovenous malformation. A postoperative hemorrhage occurred in only one patient. At 6-month follow-up, 84.2% of patients had good prognosis outcomes, and 13.5% had poor outcomes. Conclusion: Liquid embolization agents are effective for hemorrhagic peripheral intracranial aneurysms; however, safety depends on the subtypes. For peripheral hemorrhagic aneurysms in MMD, the vessel architecture must be carefully evaluated before embolization.

## 1. Introduction

Unlike aneurysms located at arterial trunks of the carotid or vertebrobasilar artery, peripheral intracranial aneurysms (PIAs) reside distally along cerebral arteries beyond the circle of Willis; they can be primary or arise as secondary lesions of arteriovenous malformations (AVMs), moyamoya disease (MMD), arteriovenous fistulas, infections or other cerebrovascular diseases [1,2,3]. The etiology, clinical manifestations, therapies, and outcomes of hemorrhagic peripheral cerebral aneurysms differ from those of common aneurysms residing at arterial trunks. Endovascular treatment of PIAs is a less invasive surgery, to some degree, sparing the patients from hazards typical of craniotomy [2].

In the present study, 38 cases of hemorrhagic PIAs treated with liquid embolization agents in our department (from January 2018 to June 2021) were retrospectively analyzed. The data of clinical presentations, imaging characteristics, endovascular treatment strategies, and corresponding prognosis were analyzed, hoping to offer insights into the treatment of PIAs.

## 2. Methods

### 2.1. Patients

We retrospectively analyzed 38 patients admitted to Nanjing Drum Tower Hospital (January 2018 to January 2022) with hemorrhagic PIAs, treated with liquid embolic agents (LEA). Inclusion criteria were as follows: cerebral hemorrhage was confirmed by computed tomography (CT) on admission; CT angiography (CTA), digital subtraction angiography (DSA), or magnetic resonance angiography (MRA) confirmed that the lesion was a peripheral intracranial aneurysm; the hemorrhagic site was consistent with the anatomical location of the aneurysm; the aneurysm was occluded with LEA. Exclusion criteria were as follows: infected aneurysms; dysfunction of heart, kidney, or other important organs; and peripheral intracranial aneurysm clipped in open surgery.

### 2.2. Imaging

Cerebral CT, CTA, or MRA was performed to identify brain hemorrhage and related vascular diseases on admission. Whole-brain DSA was performed using the Philips Allura Xper FD20/20 biplane neuro X-ray system. Additional three-dimensional angiography was performed to scan the parent vessels, assessing vascular morphology and the aneurysm characteristics.

### 2.3. Therapeutic Strategies

After discussion based on DSA, treatment strategies were as follows: (1) For patients with hemorrhagic simple peripheral aneurysms, an LEA was used to occlude the aneurysmal sac and the parent artery. (2) For patients with MMD and peripheral hemorrhagic aneurysms, micro-catheters were carefully introduced to perform super-selective angiography, aiming to occlude the aneurysmal sacs while sparing the smoky vessels as much as possible. If the microcatheter could only reach the proximal of the parent artery, the parent artery was occluded proximally with embolic agent diffusion into the aneurysm sac maximally. (3) For cerebral AVMs with hemorrhagic peripheral (flow-related) aneurysms, complete occlusion was the primary goal, while considering AVM occlusion. Individual therapeutic strategies were based on related assessments. If subsequent microsurgery resection is essential, in addition to the embolization of the aneurysm, meningeal feeding branches and those deep supplying branches were completely occluded. If the nidus could be only partially occluded, aneurysms within AVM were the primary focus, paying attention to hemorrhagic risk factors (supplying arteries and draining veins). Depending on the circumstances, patients with perioperative hydrocephalus were treated with external ventricular drainage or ventriculoperitoneal shunt. Craniotomy was performed when necessary.

### 2.4. Procedure

The microcatheter was introduced into the aneurysmal sacs via a micro-wire under a road map after super-selective catheterization for a detailed vascular morphology assessment, focusing on the characteristics of the target aneurysm, the parent artery, the structure of AVM, and the compensation of smoky vessels. Aneurysms or AVMs were then occluded by injection of Onyx18 or Glubran-2 at the appropriate concentrations. All procedures were operated by the same neurosurgeon group. Multi-angle two-dimensional angiography combined with XPer CT was performed amid the operation to rule out brain hemorrhage.

### 2.5. Evaluation of Embolization Degree

The occlusion degree of target aneurysms, parent arteries and the accompanying AVMs were thoroughly evaluated. According to the angiography immediately after embolization, modified Raymond Roy grading [4] was used for the classification standard of aneurysm embolization: Class I, complete obliteration; Class II, neck residual; Class III, incomplete embolization (i.e., residual aneurysm). The degree of AVM embolization was classified as complete or incomplete (varying degrees of residual AVM) [5].

### 2.6. Follow-Up

Attending physicians were assigned to follow-ups 6 months postoperatively via outpatient or telephone, and DSA image follow-up was performed 6 months after the operation. The treatment effect was evaluated according to the embolization degree of the aneurysm and AVM. The modified Rankin scale (mRS) was used to evaluate the outcomes of patients: scores of 0–2 stand for good outcomes, 3–6 represent poor outcomes [6].

## 3. Results

### 3.1. Demographics and Clinical Features

We included 38 patients (20 males and 18 females), aged 18 to 77 (47.4 ± 17.8, mean ± SD). Headache was the most common symptom (78.9%), followed by vomiting (23.7%) and disturbance of consciousness (23.7%). There were ten simple PIAs, six MMD with PIAs, and 22 AVM with PIAs. The posterior circulation accounted for the most significant proportion (57.9%), followed by anterior circulation (21.1%) and intranidal aneurysms (21.1%) (Table 1).

### 3.2. Treatment Outcomes

The parent artery and simple PIAs were occluded entirely in all ten cases (class I, 100%), and two demonstrated new neurological dysfunction (one with a cerebellar infarction and other died of acute hydrocephalus caused by cerebellar infarction secondary to an indwelling microcatheter). Embolization was performed in all six PIAs with MMD (class I, 100%). Intraoperative complications occurred in two patients. One had intraoperative bleeding during the super-selective catheterization. The other (critically ill) patient was occluded at the proximal end of the parent artery resulted by the failure of microcatheter super-selection into the far-end; the patient developed a severe postoperative cerebral infarction and died of cerebral hernia despite bilateral intraventricular external drainage. Postoperative infarction occurred in three cases, two of whom developed related neurological defects. Of the 22 AVMs accompanied with PIAs, the target aneurysms were successfully embolized in 21 (class I, 95.5%) cases and incomplete embolization in one case (class II, 4.5%). The AVM nidus was completely embolized in 14 patients (63.7%) and partially embolized in eight patients (36.3%) (after aneurysmal embolization, four patients were followed up and the other four chose surgical resection of AVM).

One patient with an AVM developed an intraoperative hemorrhage and microcatheter extubation failure. Postoperative complications occurred in three cases. One patient with an AVM underwent microsurgical resection after embolization of an aneurysm in the supplying artery and died due to cerebral edema. One patient developed acute hydrocephalus after complete embolization of aneurysm and AVM nidus, which was managed with the timely placement of a ventriculoperitoneal shunt; however, the patient died suddenly four hours after shunt placement for unknown reasons. Another patient had transient left extremity adynamia and motor aphasia following complete embolization of the nidus with glue overflow. In the review of all the 38 cases, intraoperative hemorrhage occurred in four patients (10.52%), intraoperative thrombosis took place in one patient (2.6%), postoperative cerebral infarction occurred in four patients (10.52%), two patients (5.3%) suffered microcatheter retention and postoperative hemorrhage occurred in one patient (2.6%). Details are shown in Table 2.

### 3.3. Follow-Up

Of the 34 discharged patients, 97.1% (33/34) completed follow-up, with one lost to follow-up. Except for one patient who refused DSA, 97% (32/33) of patients completed follow-up angiography without recurrence. According to the follow-up result based on the modified Rankin scale at 6 months, 84.2% of the patients had good outcomes, and 13.5% had poor outcomes (Table 2).

## 4. Discussion

PIAs reside in cerebral vascular terminals or peripheral branches and are idiopathic or derived from MMD, cerebral AVM, arteriovenous fistula, or other cerebrovascular diseases [7]. Guidelines for PIA management have not been established. Spontaneous hemorrhagic simple PIAs and secondary PIAs share little similarity in therapeutic concepts or methods [8]. In the present study, we presented the results of liquid embolization to treat PIA and provide a model for our colleagues.

### 4.1. Simple PIAs

Although saccular aneurysms comprise most PIAs, those with other morphologies (fusiform, dissecting, or pseudoaneurysm) are not rare [9], and they share morphologies with infectious intracranial aneurysms secondary to infective endocarditis. Infectious intracranial aneurysms are often multiple and carry a high risk of rupture. The aneurysmal neck is often ambiguous, and aneurysm morphology varies; new aneurysms could occur in a short period [10].

The paths reaching PIAs are always fine and tortuous during intravascular therapy, complicating the super selection; therefore, thorough pre-operative assessment is essential [11]. There are three types of microcatheters for delivering stents, coils, or LEA. The microcatheter delivering the LEA can bypass tortuous vessels. The therapeutic strategies for treating simple PIAs include aneurysmal occlusion with or without parent artery preservation. The risk of hemorrhage and aneurysm recurrence is high even if the parent arteries have been preserved [12,13]. Hence, simultaneous aneurysm and parent artery embolization is preferred for simple saccular or fusiform PIAs. In the present study, fusiform or dissecting aneurysms predominated. The aneurysms with parent arteries were occluded using an LEA.

The most commonly used LEAs are Glubran glue and Onyx; both possess excellent diffusion [14]. Even if the microcatheter could not reach aneurysmal sacs, gels can diffuse to the aneurysm sacs, avoiding the risk of rebleeding under the occlusion of parent arteries. However, once diffused into the distal end of parent arteries, gels can damage the countercurrent blood flow from pial branches, which may further lead to cerebral infarction. Furthermore, catheter withdrawal failure might occur if gels diffused toward the proximal end of the parent artery. Therefore, coiling could serve as a supplement when necessary.

It is critical to determine whether the parent artery supplies the functional areas [11]. Cerebral vascular fragments distal to P2 or P3 of the posterior cerebral artery [15], the internal auditory canal segment of the anterior inferior cerebral artery (AICA), and the tonsil of the posterior inferior cerebral artery (PICA) are safe for occlusion [16]. AICA occlusion is acceptable based on its abundant collateral circulations. In the present study, of the 10 patients with simple PIAs, catheter withdrawal failure occurred in one patient with a PICA aneurysm. The patient died of acute hydrocephalus 36 h after the embolization, although external ventricular drainage was performed. The distal aneurysm at the right PICA was occluded in one patient with coils and Onyx.

### 4.2. Moyamoya Disease with PIAs

Patients with MMD usually present with intracranial hemorrhage or cerebral ischemia. Concurrent PIA rupture is a critical reason for intracranial hemorrhage in patients with MMD [17]. MMD with intracranial aneurysms could be classified into Type 1, aneurysms arising from major arteries near the circle of Willis and Type 2, peripheral aneurysms originating from vessel dissection or dilation (MMD with PIAs) [18]. The treatment of the Type 1 aneurysms is like that of common aneurysms; however, the potential risk of antiplatelet-therapy-related hemorrhage of Type 1 should be assessed if stents are needed [19]. PIAs in MMD are likely to be missed [20] because the smoky arterial imaging often interferes with identifying the aneurysm sac based on CTA, MRA, or DSA. Thus, for hemorrhagic MMD, three-dimensional vascular structural reconstruction and special-angle angiography should be performed when necessary.

After reviewing 275 patients with 313 PIAs with MMD, Anthony et al. found that 95% of patients treated with endovascular therapy had no or minimal deficit, in contrast to open surgery (69.6%) [21]. The self-healing of PIAs of MMD patients was reported during follow-up [22] or after extracranial–intracranial bypass [19]. Therefore, endovascular treatment for PIAs in MMD should be based on comparing the risk of rebleeding with follow-up or extracranial–intracranial bypass [19]. In our study, there was a dissecting aneurysm located at the distal smoky middle cerebral artery, inaccessible by the microcatheters, which were occluded from the proximal end of its parent arteries, resulting in bilateral hemisphere infarction and herniation (Figure 1). Therefore, we believe that aneurysm occlusion or parent artery occlusion could be performed only if compensatory arteries are small-scale or non-functional, as confirmed by the microcatheter super-selective angiography [23]. Otherwise, extracranial–intracranial bypass should be considered [18].

### 4.3. AVM with PIAs

AVMs with aneurysms comprise around 20% of all intracranial AVMs, and 50% of AVMs had intracranial hemorrhage, of 49% were due to accompanying aneurysm rupture [24]. Redekop et al. classified aneurysms as Type I, aneurysms located within nidus; Type II, flow-related aneurysms further divided into Type IIa (aneurysms located close to the proximal of the supplying artery, also known as the trunk type) and Type IIb (aneurysms close to the nidus and associated with the flow of the nidus), and Type III, aneurysms that are not related to an AVM [25].

In the present study, all three subtypes of aneurysms were included. The hemorrhagic risk of AVM with PIAs (Type I or IIb) was 6–16% [26]. Ruptured PIAs bear even higher hemorrhagic risks [27]; active treatment strategies should be adopted. This finding is consistent with our results and experience. For Type I, super-selection with the microcatheter is critical. Once the microcatheter approaches the aneurysms, it is safe to simultaneously occlude the aneurysms and nidus with an LEA when venous drainage is clear. For type IIb aneurysms, given the higher embolization risk of type IIb aneurysms, thorough pre-operative assessments are necessary [24]. These determinations must include (1) whether the aneurysms are ruptured; (2) the risk of hemorrhage; (3) the consequences of parent arteries occlusion; (4) the priority of the occlusion of AVM and aneurysms; and (5) the safety of microcatheter withdrawal and the necessity of utilization of detachable catheters.

In this study, Type I aneurysms (17, 77.3%) were wholly embolized in all patients, and the AVM nidus in four was partially occluded, with two choosing follow-up and two undergoing microsurgical resection. Two patients developed intra-operative hemorrhages during microcatheter super-selection; one had intraoperative thrombosis with effective handling. Four of the five Type IIb patients had complete occlusions of the aneurysm; however, the pericallosal artery branch aneurysm remained visible after the partial occlusion in one patient. One Type IIb patient had a curative occlusion of AVM, two had partial occlusions for further follow-up, and the other two patients had partial occlusions of the main supplying arteries for further surgical resection. No operation related complications occurred. According to our experience, the application of an Apollo detachable microcatheter is recommended for aneurysms that reside within three centimeters of an AVM nidus. During the procedure, the tip of the detachable microcatheter should be positioned close to the nidus with the detachment point proximal to the aneurysm, allowing the gel reflux to aneurysmal sacs to occlude the aneurysms and nidus together (Figure 2). If two microcatheters could be accommodated within the parent artery simultaneously, detachable microcatheters are recommended for injecting the LEA after aneurysm coiling via the other microcatheter. Furthermore, the use of detachable microcatheters could decrease the risk of catheter withdrawal. In addition, if the parent artery irrigates essential functional areas, it is critical to preserve the blood flow, manage accompanying aneurysms, and consider coiling or microsurgery [1].

In conclusion, LEAs are satisfactory for PIA treatments. We believe that for simple PIAs and AVM accompanying PIAs, the safety and efficacy of liquid embolization are quite acceptable. However, given the chronic progressive characteristics of MMD, the concurrent aneurysms might change due to intracranial hemodynamic fluctuations. Their aneurysmal cavities might be accompanied by extensive compensatory vessels, and the parent arteries are always not easy to super-select; therefore, comprehensive clinical studies are warranted to develop therapies for MMD with PIAs.

## Figures and Tables

**Figure 1 brainsci-12-01264-f001:**
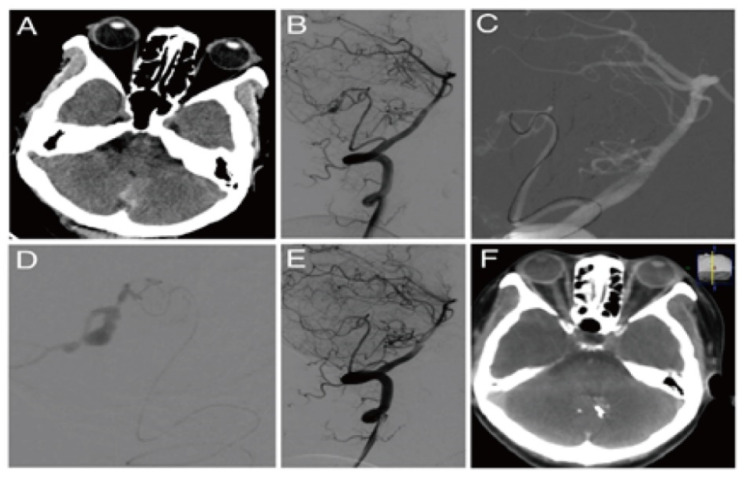
Moyamoya disease with a peripheral hemorrhagic aneurysm. (**A**) Preoperative CT revealed periventricular and ventricular hemorrhage. (**B**) Left internal carotid arteriography revealed smoky vascular hyperplasia with a false aneurysm at the end of the lenticulostriate artery. (**C**) The parent vessels were too tortuous and slender for super-selection. A Marathon microcatheter was located at the beginning of the parent artery. (**D**) Glubran was used to block the parent artery with satisfactory diffusion to the distal aneurysm cavity. (**E**) After the embolization, the aneurysm and parent artery disappeared on arteriography. (**F**) Postoperative CT revealed that the position of the glue coincided with the location of a brain hemorrhage.

**Figure 2 brainsci-12-01264-f002:**
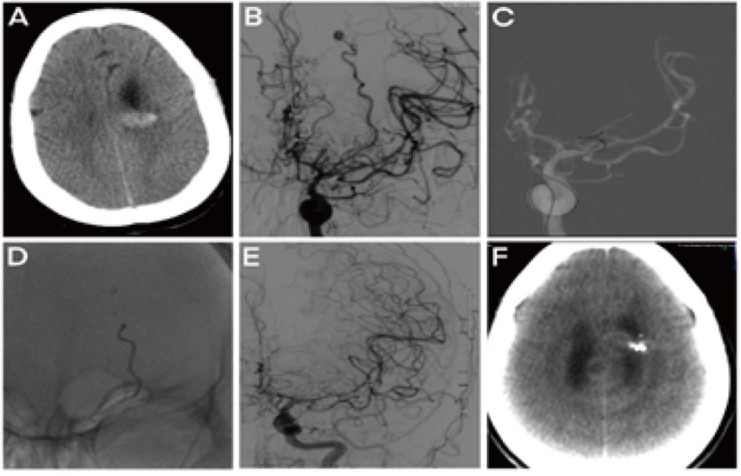
AVM with peripheral hemorrhagic aneurysm. (**A**) Preoperative CT revealed hemorrhage in the corpus callosum’s ventricle and splenium (**B**) DSA demonstration of the splenium of the corpus callosum AVM supplied by the right posterior cerebral artery branches and the branches of the right anterior cerebral artery, accompanied by a blood flow related aneurysm. (**C**) Super selection and positioning of the microcatheter through the left vertebral artery, preparing for the embolization. (**D**) Embolization of the AVM and accompanied aneurysm with Onyx. (**E**) Arteriography after embolization revealing the complete embolization of the AVM and the aneurysm. (**F**) CT scan before discharge, revealing adequate treatment.

**Table 1 brainsci-12-01264-t001:** Demographics and clinical features.

	Simple PIAs	AVM	MMD	Total
**Number of patients**	10	22	6	38
**Age**	59.7 ± 9.8 years	40.8 ± 18.4 years	48.5 ± 7.0 years	47.4 ± 17.8 years
**Gender**				
Male	6 (60.0%)	12 (54.5%)	2 (33.3%)	20 (52.6%)
Female	4 (40.0%)	10 (45.5%)	4 (66.7%)	18 (47.4%)
**Manifestations**				
Headache	10 (100.0%)	16 (72.7%)	4 (66.7%)	30 (78.9%)
Dizziness	3 (30.0%)	3 (13.6%)	1 (16.7%)	7 (18.4%)
Vomiting	3 (30.0%)	0	6 (100.0%)	9 (23.7%)
Loss of consciousness	3 (30.0%)	2 (9.1%)	4 (66.7%)	9 (23.7%)
Extremity adynamia	0	3 (13.6%)	1 (16.7%)	4 (10.5%)
**Aneurysm locations**				
** Anterior Circulation**		5 (22.7%)	3 (50%)	8 (21.1%)
Anterior cerebral artery	0	4 (18.2%)	0	4 (10.5%)
Middle cerebral artery	0	1 (4.5%)	2 (33.3%)	3 (7.9%)
Anterior choroidal artery	0	0	1 (16.7%)	1 (2.6%)
** Posterior Circulation**	10 (100%)	9 (40.9%)	3 (50.0%)	22 (57.9%)
Posterior cerebral artery	0	3 (13.6%)	3 (50.0%)	6 (15.8%)
Superior cerebellar artery	4 (40.0%)	3 (13.6%)	0	7 (18.4%)
Anterior inferior cerebellar artery	2 (20.0%)	0	0	2 (5.3%)
Posterior inferior cerebellar artery	4 (40.0%)	1 (4.5%)	0	5 (13.2%)
Vertebrobasilar artery	0	2 (9.1%)	0	2 (5.3%)
**Intranidal**	0	8 (36.4%)	0	8 (21.1%)

**Table 2 brainsci-12-01264-t002:** Treatment outcome and follow-up.

Treatment Results	Simple PIAs	AVM	MMD	Total
**Aneurysm Treatment**				
Curative occlusion	10 (100.0%)	21 (95.5%)	6 (100%)	37 (97.4%)
Partial occlusion	0	1 (4.5%)	0	1 (2.6%)
**Malformation Treatment**				
Complete Occlusion	\	14 (63.7%)	\	
Partial Occlusion	\	4 (18.2%)	\	
Partial Occlusion + microsurgery	\	4 (18.2%)	\	
**Complications**				
Intraoperative hemorrhage	0	1 (4.5%)	3 (50.0%)	4 (10.5%)
Intraoperative thrombosis	0	0	1 (16.7%)	1 (2.6%)
Postoperative hemorrhage	0	0	1 (16.7%)	1 (2.6%)
Catheter withdrawing failure	1 (10.0%)	0	1 (16.7%)	2 (5.2%)
Cerebral Infarction	1 (10.0%)	1 (4.5%)	2	4 (10.5%)
**Follow-up (mRS score at 6 months)**				
0–2	8 (80.0%)	19 (86.4%)	5 (83.3%)	32 (84.2%)
3–6	2 (20.0%)	2 (9.1%)	1 (16.7)	5 (13.5%)
Loss	0	1 (4.5%)	0	1 (2.6%)
**DSA Follow-up**	32 (84.2%)			32 (84.2%)

## Data Availability

Data supporting the findings in this study are available from the corresponding author upon reasonable request.

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
