# Peer review of "Efficacy of Liquid Embolic Agent Treatment in Hemorrhagic Peripheral Intracranial Aneurysms: A Single-Center Experience"

_brainsci, 2022, doi:10.3390/brainsci12091264_

Round 1

Reviewer 1 Report

In the manuscript presented by Zhuang et al., the authors assessed the efficacy of liquid embolic agents in treating three types of hemorrhagic peripheral intracranial aneurysms in a single-center retrospective study. They found that liquid embolic agents are safe and efficient in treating simple peripheral intracranial aneurysms and AVM accompanied peripheral intracranial aneurysms. For MMD with peripheral intracranial aneurysms, prudent evaluation of the angioarchitecture must be carried out before embolization. This study was well designed and these findings are interesting. However, there are a number of issues that need to be addressed before publication.

1.  Which database was used for selecting patients? According to the study, the database may be from the institute of the authors. Please state where the patients were admitted in the methods and results section. A flowchart describing the sampling procedure needs to be added.

2.  The specific standard of mRS should be listed in a table or described in detail.

3.  A table comparing clinical features, treatment outcome and mRS among these three patient categories (simple peripheral aneurysm, MMD with hemorrhagic peripheral aneurysm, and AVMs with hemorrhagic peripheral aneurysm) needs to be shown.

4.  Were the patients under the three categories operated by the same neurosurgeon or neurosurgeon group?

Author Response

Dear reviewer : 

Thank your for your thorough comments.

Reply are as follows:

1 This is a retrospective study. Patients were admitted to the neurosurgery department of our hospital. Details were added to our manuscript. Thank you for your comments. 

2 We supplemented a reference to "methods". Thank you.

3 Since the differences between various diseases( simple PIAs, AMV, MMD) were not our main objective, we did not distinguish those three diseases. However, we still update our table in our manuscript which may provide more information for readers. Thank you.  

4 Sure. All procedures involved in the present study were operated by one neurosurgeon team. We noted in our "methods" as you reminded. Thank you for your valuable comments. 

Kind regards

Reviewer 2 Report

This is a retrospective study of 38 patients who had intracranial aneurysms treated with liquid embolic agents (Onyx19 or Glubran-2) between 2018-2022. This paper is descriptive and outlines the different complications that patients experience after the procedure. Of interest, the authors classify the aneurysms as simple PIAs, MMD with PIA, and AVM with PIAs. It was interesting to see that these liquid embolic agents were used on MMD patients but unfortunate that there was a MMD patient who had poor outcome. Would be interesting in further studies to characterize risk factors for poor outcomes in MMD patients treated with liquid agents.

Recommend proof-reading paper for grammatical errors. Some of these are:

-Please add PIA to abbreviation list

- In abstract, please fully spell out MMD and AVM before using these abbreviations.

-In introduction, first sentence, please use "Circle of Willis" instead of Willis circle. In that same sentence, please reconsider the word "solitary", do you mean to say "primary?"

-In introduction, last sentence of first paragraph, please reword to make it more clear. Reconsider saying "provide attractive minimally invasion". Could say "provide a minimally invasive alternative, sparing the patients...."

-In section of 2.3, therapeutic strategies, reconsider the use of phrase "occluded emphatically", I think you meant to say something else. 

Author Response

Dear reviewer: 

Thank you for your thorough comments.

After reading your careful comments, we sought for professional language editing service. The certificate are attached below. 

Thanks again.

Round 2

Reviewer 1 Report

I think the revised manuscript meets the publishing requirments and I suggest acception.

Author Response

Thank you for time and patience for the manuscript. Your comments are quite valuable and helpful for it. Thanks again!